# Constituents of *Aquilaria sinensis* Leaves Upregulate the Expression of Matrix Metalloproteases 2 and 9

**DOI:** 10.3390/molecules26092537

**Published:** 2021-04-26

**Authors:** Sui-Wen Hsiao, Yu-Chin Wu, Hui-Ching Mei, Yu-Hsin Chen, George Hsiao, Ching-Kuo Lee

**Affiliations:** 1Ph.D. Program in Drug Discovery and Development Industry, College of Pharmacy, Taipei Medical University, 250 Wu Xin Street, Taipei 11031, Taiwan; suifeng0506@gmail.com; 2Graduate Institute of Pharmacognosy, Taipei Medical University, 250 Wu Xin Street, Taipei 11031, Taiwan; tmc761038@tmu.edu.tw; 3Department of Science Education, National Taipei University of Education, Taipei 10671, Taiwan; hcmei@tea.ntue.edu.tw; 4Taichung District Agricultural Research and Extension Station, Council of Agriculture, Executive Yuan, Taichung 42081, Taiwan; ychen@tdais.gov.tw; 5Department of Pharmacology, Taipei Medical University, 250 Wu Xin Street, Taipei 110, Taiwan; 6School of Pharmacy, Taipei Medical University, 250 Wu Xin Street, Taipei 110, Taiwan; 7Department of Chemistry, Chung Yuan Christian University, Zhongbei Road, Zhongli District, Taoyuan City 320314, Taiwan

**Keywords:** *Aquilaria sinensis*, pheophorbide A, MMP-2, MMP-9, HT-1080

## Abstract

In this novel study, we isolated 28 compounds from the leaves of *Aquilaria sinensis* (Lour.) Gilg based on a bioassay-guided procedure and also discovered the possible matrix metalloprotease 2 (MMP-2) and 9 (MMP-9) modulatory effect of pheophorbide A (PA). To evaluate the regulatory activity on MMP-2 and MMP-9, the HT-1080 human fibrosarcoma cells were treated with various concentrations of extracted materials and isolated compounds. PA was extracted by methanol from the leaves of *A. sinensis* and separated from the fraction of the partitioned ethyl acetate layer. PA is believed to be an active component for MMP expression since it exhibited significant stimulation on MMP-2 and proMMP-9 activity. When treating with 50 μM of PA, the expression of MMP-2 and MMP-9 were increased 1.9-fold and 2.3-fold, respectively. PA also exhibited no cytotoxicity against HT-1080 cells when the cell viability was monitored. Furthermore, no significant MMP activity was observed when five PA analogues were evaluated. This study is the first to demonstrate that C-17 of PA is the deciding factor in determining the bioactivity of the compound. The MMP-2 and proMMP-9 modulatory activity of PA indicate its potential applications for reducing scar formation and comparative medical purposes.

## 1. Introduction

*Aquilaria sinensis* (Lour.) Gilg is an evergreen woody plant with high economic value that is widely distributed in the tropical area of China. Possessing analgesic, sedative, and antiemetic effects, the resinous heartwood of *A. sinensis*, which is the major source of agarwood in China, has been used as traditional medicine for centuries [1]. Apart from the agarwood, the non-medical parts of leaves and flowers are consumed as a healthy herbal tea as well. The bioactivity of agarwood has been highly valued for years, while studies on the leaves have still been limited. The major metabolic components of *A. sinensis* are polysaccharides, amino acids, flavonoids and their glycosides, phenols, and xanthones [2,3,4,5]. Previous studies have reported anti-inflammatory and analgesic activity [6,7], laxative activity [8], an α-glucosidase inhibitory effect [2], a nitric oxide inhibitory effect [9], anticancer activity [10,11], and blood glucose regulatory effects [12] of the isolated compounds and extracts of *A. sinensis* leaves. Given the abundance and availability of leaves, there can potentially be a variety of pharmacological applications of the leaves from *A. sinensis.*

Matrix metalloproteinases (MMPs) are a family of enzymes important in the regulation of developmental and homeostatic remodeling of the extracellular matrix (ECM). Family members of MMPs can proteolytically degrade all components of the ECM and are subdivided into collagenases (MMP-1, -8, -13), gelatinases (MMP-2, -9), stromelysins (MMP-3, -10, -11), matrilysin (MMP-7), elastase (MMP-12), and MT-MMPs (MMP-14, -15, -16, -17) [13]. The MMP-2 and MMP-9, also known as gelatinase A and gelatinase B, respectively, share some of the same substrates, such as gelatins, collagens IV and V, aggrecan, elastin, and vitronectin [14]. MMP-2 can also degrade collagens I, VII, X, XI, β-amyloid protein precursor, and other substrates [14]. To investigate the effect of components isolated from *A. sinensis* on MMPs expression activity, the widely used HT-1080 cell model was used in this study. The HT1080 cell line is derived from a human fibrosarcoma to mimic stromal fibroblasts and is sustained to express MMP-2 and MMP-9 [15,16].

In the present study, the phytochemical properties of the components isolated from *A. sinensis* leaves and their regulatory effects on MMP-2 and MMP-9 expression were investigated.

## 2. Results

### 2.1. Bioassay Guided Compound Isolation from A. sinensis Leaves

In a preliminary biological evaluation, methanolic crude extracts of *A. sinensis* leaves demonstrated the ability to promote MMP-2 and MMP-9 activity at a concentration of 100 μg/mL. An investigation of the active principles of this plant was thus undertaken by using a bioassay guided method. After being partitioned into water (91.34 g), *n*-butanol (77.58 g), and ethyl acetate (EA) layers (225.27 g), the effects of these layers (100 μg/mL) on MMPs expression were evaluated by gelatin zymography. The EA layer was the most potent (Figure 1).

In order to find the constituents of *A. sinensis* leaves that regulate MMP-2 and MMP-9 activity, we carried out a biologically guided purification strategy. The EA layer in the methanol (MeOH) extract of *A. sinensis* leaves showed potential effects. The subsequent separation and identification of biologically active ingredients were thus focused on this layer. The most effective fractions were further fractionated and purified by repeated chromatography on silica gel columns and semi-preparative HPLC with refractive index detector to obtain twenty-eight compounds (Figure 2). The representative HPLC trace of compound AQ20 is shown in Figure 3.

### 2.2. Compounds Isolated from the Ethyl Acetate Layer of of Aquilaria sinensis Leaves

To analyze the structures of the compounds separated from various fractions of the EA layer, the spectra from NMR and MS spectroscopic analysis were compared with the previously reported data. The classifications for identified isolated compounds were recognized as:

Terpenoids: loliolide (AQ4) [17], phytol (AQ7) [18], squalene (AQ15) [19]; Ligand: syringaresinol (AQ12) [20]; Phenolics: *p-*hydroxybenzoic acid (AQ8) [18], β*-*truxinic acid (AQ10) [21], *p-*hydroxyphenethyl *trans*-ferulate (AQ13) [22]; Lipids: methyl (6*Z*,9*Z*,12*Z*)-octadecatrienoate (AQ5) [23], palmitic acid and oleic acid (AQ6) [24,25], methyl-2-hydroxypentanedioate (AQ17) [26], glyceryl -9*Z*-octadecenate (AQ21) [27], glyceryl 1-monononadecylate (AQ22) [28], 1-[nonadeca-(9Z,12Z)-dienoyl]-sn-glycerol (AQ23) [29]; Chromones: 7-hydroxy-6-methoxy-2-[2-(4′-methoxyphenyl)ethyl] chromone (AQ11) [30], 6-hydroxy-7-methoxy-2-[2-(3′-hydroxy-4′ -methoxyphenyl)ethyl] chromone (AQ16) [30]; Flavonoids: genkwanin (AQ1) [31], apigenin (AQ2) [32], velutin and pilloin (AQ3) [33,34];

Glycosides: 2-phenylethyl β-glucopyranoside (AQ14) [35], benzyl β-glucopyranoside (AQ18) [36], iriflophenone 2-*O*-α-rhamnoside (AQ19) [8], 1-*O*-caprylyl-2-*O*-linolenoyl-3-*O*-(β-d-glucopyranosyl)- *rac*-glycerol (AQ24) [37], 1-*O*-caprylyl-2-*O*-linoleoyl-3-*O*-(β-d-glucopyranosyl)- *rac*-glycerol (AQ25) [37], 1,2-*O*-dimyristoyl-3-*O*-(β-d-glucopyranosyl)-*rac*- glycerol (AQ26) [37]; Pheophorbides: methyl pheophorbide A (AQ9) [38], and pheophorbide A (PA) (AQ20) (Appendix A) [39].

### 2.3. Evaluation of Regulatory Effects on MMP-2 and MMP-9 Expression in HT-1080 Cells

The MMP-2 and MMP-9 modulating activity of the pure isolated compounds were measured at the concentrations of 50 μM in comparison with blank groups. Gelatin zymography was performed in triplicate, and the data were obtained from independent zymogram. As shown in Figure 4, the compound AQ20, which was identified as PA, distinctly increased the expression of MMP-2 and proMMP-9 by 1.9-fold and 2.3-fold elevation, respectively, with a 99.9% confidence level. While the protein levels of all other isolated compounds showed no significant difference between that of blank. To determine the cell toxicity of PA, HT-1080 cells were subjected to MTT cell viability assay in the presence of 50 μM PA. The cell viability of the HT-1080 cells was 94%, indicating that PA had no cell toxicity. With the exception of compound AQ3 where viability was 65%, all other compounds showed more than 75% cell viability.

Numerous concentrations were used to further investigate the dose–response effect of PA (Table 1 and Figure 5). The data on gelatin zymography were obtained from independent zymogram. For MMP-9, the expression levels were significantly elevated (*p* < 0.001) at concentrations of 25 and 50 μM. For MMP-2, the stimulating effect was observed at all of the tested concentrations. The *p*-value was lower than 0.05 at the concentration of 2.5 μM and lower than 0.001 at concentrations from 5 to 50 μM. Positive dose–response relationships between PA and the MMPs were clarified since higher expression levels were obtained along with increased concentrations of PA.

Along with the elevating effects of PA on MMP-2 and MMP-9 activity, the effects on MMP activity of five analogues of PA—bidenphytins A (20A), 13^2^-hydroxypheophytin b (20B), (13^2^S)-13^2^-hydroxypheophytin a (20C), bidenphytins B (20D), and (13^2^R)-13^2^-hydroxypheophytin a (20E) (Figure 6) [40]—were also measured. The tested PA analogues were isolated in lab from leaves of *Biden pilosa* in previous study. However, the MMP-2 and MMP-9 expression levels showed no significant differences between each of the five PA analogues and blank, indicating that the five analogues had no MMP regulatory activity (Appendix A).

## 3. Discussion

In this study, 28 compounds were isolated from the leaves of *A. sinensis*, including three terpenoids, one ligand, three phenolics, seven lipids, two chromones, four flavonoids, six glycosides, and two pheophorbides. Among the 28 isolated pure compounds, PA presented effective promoting activity on MMP-2 and MMP-9.

Increased expression of MMP-2 and MMP-9 was reported to induce tumor angiogenesis, inflammation, and cancer metastasis [41]. Additionally, both MMP-2 and MMP-9 participate in the process of wound repair and tissue remodeling [42]. Prolonged and sustained MMP-2 expression during dermal wound repair suggested that MMP-2 may play a role in granulation and early remodeling phases of repair [43]. Upregulation of MMP-9 expression in the regeneration phase of wound healing can possibly prevent scar formation [44]. The MMP-2 and MMP-9 upregulatory activity revealed the potential scar reducing and comparative medical applications of PA.

PA is a metabolic breakdown product of chlorophyII in plants. Until now, various bioactivities of PA have been reported, including anticancer, antiviral, anti-inflammatory, antioxidant, immunostimulatory, and anti-parasite activity [45]. PA also serves as a good photosensitizer that absorbs energy from light to generate reactive oxygen species [46]. The regulatory effect of PA on MMP-2 and MMP-9 expression has also been reported. In ultraviolet (UV) B-exposed CCD-986sk human fibroblasts, PA and PA-derivatives including pyroPA and PA methyl ester reduced UVB-induced MMP-1 secretion and mRNA expression of MMP-1, -2, and -9 [47]. In the present study, expression of MMP-2 and MMP-9 were upregulated by PA in HT-1080 cells. The opposite result may due to the effect of UV irradiation on PA.

In contrast with PA, the five analogues showed no effect on MMP-2 and MMP-9 expression. In addition, the compound AQ9, identified as methyl PA, did not induce significant activity either. This can potentially be attributed to the phytol group of the analogues and the methyl group on C-17 of AQ9. Based on structure–activity relationships, the structure of the substituent on C-17 was suggested to be a crucial factor influencing the activity on MMP-2 and MMP-9. This result could potentially contribute to further investigations of the bioactivity of compounds similar in structure to PA.

## 4. Materials and Methods

### 4.1. General

Optical rotations were measured on a JASCO P-1020 digital polarimeter (Tokyo, Japan). ^1^H- and ^13^C-NMR were acquired on a Bruker DRX-500 SB spectrometer (Ettlingen, Germany). High-resolution and low-resolution mass spectra were obtained using an ABI Q-Star XLQ-TOF (Applied Biosystems, Foster, CA, USA) and ABI API 4000 Q-TRAP (Foster City, CA, USA), respectively. IR spectra were recorded on a JASCO FT/IR 4100 spectrometer (Tokyo, Japan). UV spectra were measured on a Thermo Helios μ spectrophotometer (Bellefonte, CA, USA). Silica gel (40–63 μm, Merck, Darmstadt, Germany) was used for gravity column chromatography. Pre-coated silica gel plates (Si 60 F_254_, 0.2 mm, Merck, Darmstadt, Germany) were used for analytical TLC. Semi-preparative HPLC was performed on a Hitachi L-7110 HPLC with a refractive index detector (Thermo Separation Products, Sunnyvale, CA, USA). A Phenomenex^®^ Luna silica column (5 μm, 10 × 250 mm, Phenomenex, CA, USA) and a VP 250/10 NUCLEODUR C18 HTec column (5 μm, 10 × 250 mm, MACHEREY-NAGEL, Düren, Germany) were used.

### 4.2. Plant Material

The leaves of *A. sinensis* were collected from Changhua County in June 2002 and were identified by Dr. S. Y. Chen at the Council of Agriculture, Executive Yuan. Voucher Specimens (No. CKL06012002) were deposited at the School of Pharmacy, Taipei Medical University, Taipei, Taiwan.

### 4.3. Extraction and Isolation

The dried leaves (2.9 kg) of *A. sinensis* were extracted three times with 20 L of MeOH, then partitioned between EA and water (1:1, *v/v*). Subsequently, the dried EA layer (190 g) was pre-coated using 250 g silica gel (70–230 mesh) applied onto a 2500 g silica gel open column (230–400 mesh) and eluted with mixtures of *n*-hexane (*n*-Hex), EA, and MeOH in a stepwise gradient mode. Ten conditions were used for elution and every 1000 mL of eluent was collected in one bottle. Eventually, 164 bottles were collected and combined into 10 fractions based on the elution condition. Each bottle of collected eluent was checked for its composition by TLC dipped in 10% H_2_SO_4_ in ethanol for the basis of further normal-phase or reversed-phase semi-preparative HPLC analysis. Purity of the compounds was checked by NMR and was higher than 80%. The structure of the compound was identified by NMR and MS.

The fraction 2 was eluted with *n*-Hex/EA (95:5, *v/v*) and 20 bottles were collected. The eluents were purified by HPLC on a semi-preparative normal-phase column at a flow rate of 3 mL/min and afforded **5** (59.0 mg, RT 22.2 min with *n*-Hex/EA (99:1, *v/v*))**, 6a** + **6b** (323.4 mg, RT 18.6 min with *n*-Hex/EA (90:10, *v/v*))**, 7** (57.2 mg, RT 27.9 min with *n*-Hex/EA (90:10, *v/v*)), and **15** (249.3 mg, RT 6.13 min with *n*-Hex/EA (99:1, *v/v*)).

The fraction 3 was eluted with *n*-Hex/EA (90:10, *v/v*) and 20 bottles were collected. The eluents were purified by HPLC on a semi-preparative normal-phase column at a flow rate of 3 mL/min and afforded **20** (19.3 mg, RT 12.9 min with *n*-Hex/EA/acetone (80:10:1, *v/v/v*)).

The fraction 4 was eluted with *n*-Hex/EA (80:20, *v/v*) and 20 bottles were collected. The eluents were purified by HPLC on a semi-preparative normal-phase column at a flow rate of 3 mL/min and afforded **8** (47.7 mg, RT 12.2 min with *n*-Hex/EA (70:30, *v/v*)) and **9** (13.5 mg, RT 21.7 min with *n*-Hex/EA (70:30, *v/v*)).

The fraction 5 was eluted with *n*-Hex/EA (70:30, *v/v*) and 19 bottles were collected. The eluents were purified by HPLC on a semi-preparative normal-phase column at a flow rate of 3 mL/min and afforded **1** (12.2 mg, RT 14.6 min with *n*-Hex/EA (60:40, *v/v*))**, 2** (21.8 mg, RT 19.9 min with *n*-Hex/EA (60:40, *v/v*))**, 3a** + **3b** (56.8 mg, RT 21.9 min with *n*-Hex/EA (60:40, *v/v*))**, 4** (62.0 mg, RT 20.8 min with *n*-Hex/EA (50:50, *v/v*))**, 11** (17.4 mg, RT 26.2 min with *n*-Hex/EA (30:70, *v/v*))**, 21** (157.1 mg, RT 67.2 min with *n*-Hex/EA/acetone (71:28:1, *v/v/v*))**, 22** (64.2 mg, RT 72.1 min with *n*-Hex/EA/acetone (71:28:1, *v/v/v*)), and **23** (206.0 mg, RT 77.7 min with *n*-Hex/EA/acetone (71:28:1, *v/v/v*)).

The fraction 6 was eluted with *n*-Hex/EA (60:40, *v/v*) and 17 bottles were collected. The eluents were purified by HPLC on a semi-preparative normal-phase column at a flow rate of 3 mL/min and afforded **10** (3.7 mg, RT 8.2 min with *n*-Hex/EA (30:70, *v/v*))**, 12** (43.5 mg, RT 15.4 min with *n*-Hex/EA/acetone (64:32:4, *v/v/v*))**, 13** (6.3 mg, RT 18.3 min with *n*-Hex/EA/acetone (64:32:4, *v/v/v*)), and **16** (20.2 mg, RT 24.3 min with *n*-Hex/EA/acetone (25:68:7, *v/v/v*)).

The fraction 7 was eluted with *n*-Hex/EA (40:60, *v/v*) and 20 bottles were collected. The eluents were purified by HPLC on a semi-preparative reversed-phase column at a flow rate of 2 mL/min and afforded **24** (42.7 mg, RT 8.2 min with MeOH/ACN (40:60, *v/v*))**, 25** (39.1 mg, RT 37.8 min with MeOH/ACN (40:60, *v/v*)), and **26** (68.6 mg, RT 51.9 min with MeOH/ACN (40:60, *v/v*)).

The fraction 8 was eluted with *n*-Hex/EA (20:80, *v/v*) and 19 bottles were collected. The eluents were purified by HPLC on a semi-preparative reversed-phase column at a flow rate of 2 mL/min and afforded **17** (4.1 mg, RT 10.9 min with MeOH/H_2_O (40:60, *v/v*)).

The fraction 9 was eluted with 100% EA and 11 bottles were collected. The eluents were purified by HPLC on a semi-preparative reversed-phase column at a flow rate of 2 mL/min and afforded **14** (5.7 mg, RT 33.1 min with MeOH/ACN/H_2_O (12.5:12.6:75, *v/v/v*)) and **18** (11.8 mg, RT 24.3 min with MeOH/H_2_O (1:3, *v/v*)).

The fraction 10 was eluted with 100% MeOH and 12 bottles were collected. The eluents were purified by HPLC on a semi-preparative reversed-phase column at a flow rate of 1.5 mL/min and afforded **19** (9.6 mg, RT 29.1 min with MeOH/ACN/H_2_O (1:1:5, *v/v/v*)).

### 4.4. Cell Culture

HT1080 human fibrosarcoma cells were purchased from American Type Culture Collection (ATCC: CCL-121) and were maintained in a humidified incubator at 37 °C in 5% CO2/95% air. The cells were cultured in RPMI-1640 medium (Gibco) supplemented with 10% heat-inactivated fetal bovine serum, 100 U/mL penicillin, and 100 mg/mL streptomycin.

### 4.5. Gelatin Zymography

Gelatin zymography was used to determine the expression of MMP-2 and MMP-9 [48]. HT1080 cell suspension (5 × 10^5^ cells/mL) was seeded in 24-well cell culture plates using medium supplemented with 0.5% FBS for 24 h for cell adhesion and growth. Subsequently, the cells were treated with indicated concentrations of samples with vehicle (DMSO) as blank and phorbol 12-myristate 13-acetate (PMA, 5 μM) as positive control. After 24 h of incubation, conditioned medium was collected to analyze the activity of MMP-2 and MMP-9.

### 4.6. MTT Cell Viability Assay

HT1080 cell suspension (5 × 10^5^ cells/mL) was seeded in 24-well cell culture plates for 24 h for cell adhesion and growth. After being treated with samples and incubated for 22 h, 5.5 mg/mL MTT were added to the cells and incubated for 2 h. DMSO was added for cell lysis and OD 550 nm was detected.

## Figures and Tables

**Figure 1 molecules-26-02537-f001:**
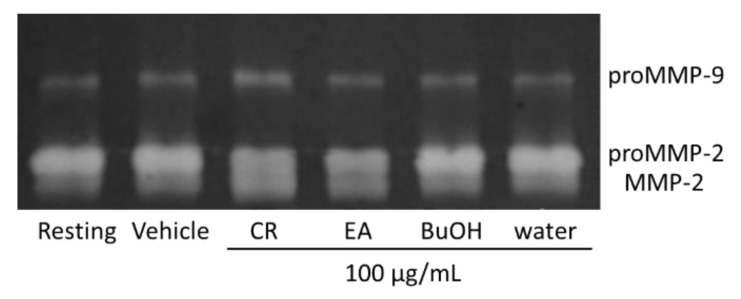
Effects of crude extracts and partitioned parts of *A. sinensis* leaves on MMP-2 and MMP-9 activity. Results of gelatin zymography. CR, crude extracts; EA, ethyl acetate layer; BuOH, butanol layer; water, water layer.

**Figure 2 molecules-26-02537-f002:**
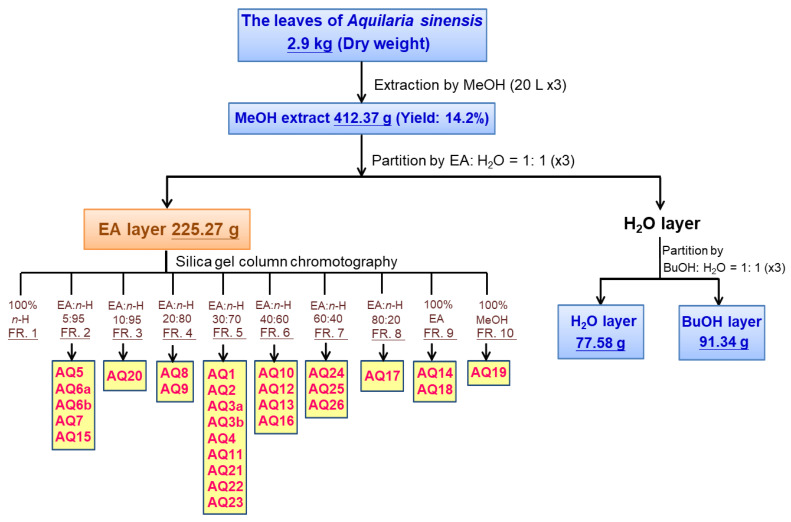
Flow diagram of bioassay-guided procedure. Bioassay-guided fractionation and isolation of the MeOH extract of the leaves of *A. sinensis* resulted in the isolation of 28 compounds.

**Figure 3 molecules-26-02537-f003:**
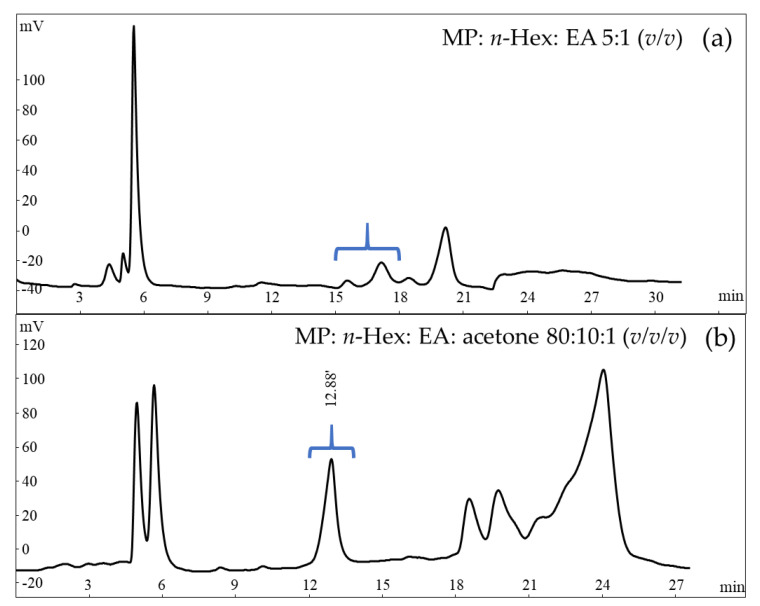
Representative HPLC trace of compound AQ20. (**a**) Eluent from RT 15-18 min was collected in one bottle and subjected to semi-preparative HPLC for further separation. (**b**) Peak at RT 12.9 min was collected and identified as AQ20. MP, mobile phase condition.

**Figure 4 molecules-26-02537-f004:**
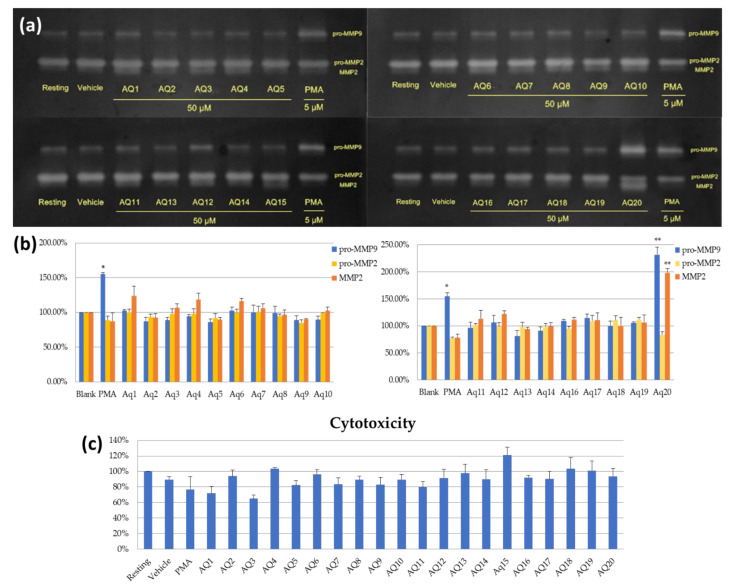
Effects of the isolated compounds AQ—AQ20 on MMP-2 and MMP-9 activity and cytotoxicity. (**a**) Results of gelatin zymography. (**b**) Activity of the compounds on MMPs expression. (**c**) Cell viability test. Data represent the mean ± SD (*n* = 3). PMA (5 μΜ); AQ1—AQ20 (50 μΜ); * *p* < 0.05, ** *p* < 0.001.

**Figure 5 molecules-26-02537-f005:**
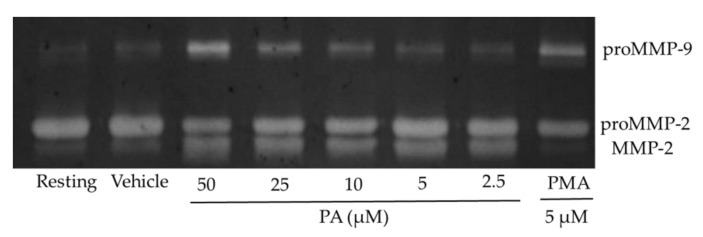
Effects of PA at various concentration.

**Figure 6 molecules-26-02537-f006:**
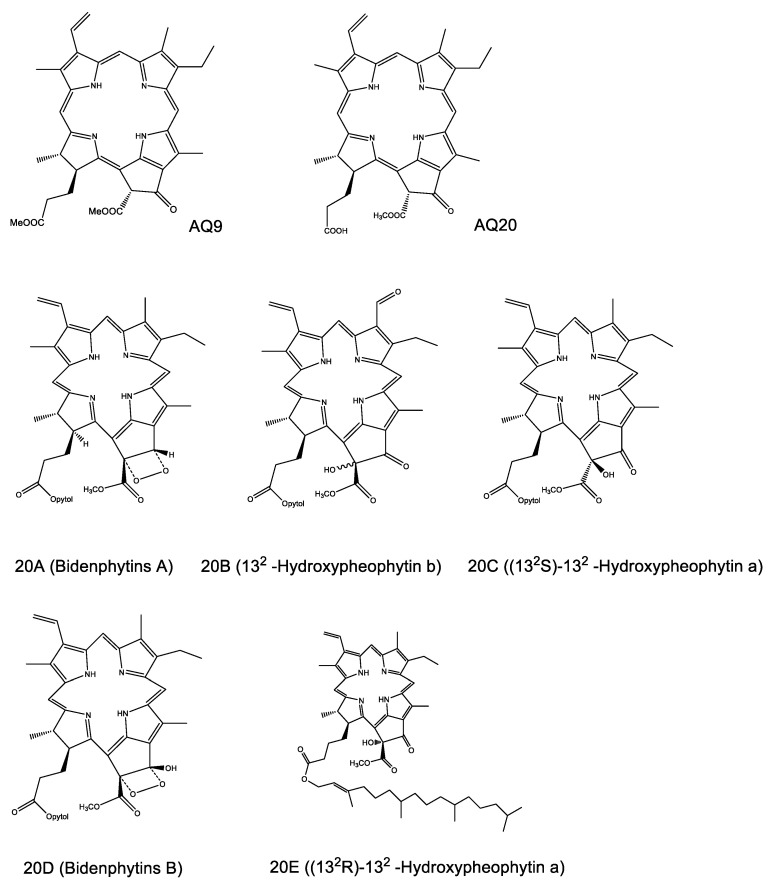
Structures of PA and its analogues. PA (AQ20), PA analogues (AQ9, 20A–E).

**Table 1 molecules-26-02537-t001:** MMP-2 and MMP-9 activation by PA at various concentrations.

Concentration (μM)	Expression Level (%) ^1^	
	pro-MMP9	MMP-2
50	230 ± 11.5 **	188 ± 6.5 **
25	146 ± 3 **	163 ± 2.5 **
10	118 ± 16	147 ± 7 **
5	104 ± 4	135 ± 2 **
2.5	104 ± 4	132 ± 11 *

^1^ The expression level was compared with blank group (100%). * *p* < 0.05, ** *p* < 0.001.

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
