# Peer review of "Constituents of Aquilaria sinensis Leaves Upregulate the Expression of Matrix Metalloproteases 2 and 9"

_molecules, 2021, doi:10.3390/molecules26092537_

Round 1

Reviewer 1 Report

The manuscript describes the separation and characterization of 28 compounds from Aquilaria sinensis and then tested them for the ability to induce MMP secretion by HT1080 cells.  This is an interesting and well structured manuscript but it requires significant modification before it could be accepted for publication. 

The results start describing the presence of MMPs but do not describe the assay – it could suggest that the MMPs were in the plant extract.  Therefore, this needs to be properly described and a short introduction of the HT1080 model needs to be added to give context to the data presented. 

In the text it is indicated that the zymogram in Figure 1 showed that ethyl acetate has the strongest impact on MMP production – this is not obvious from the figure and it appears to me that the butanol and water fractions actually show higher levels of proMMP2.  The figure requires quantitative densitometry representing multiple independent experiments. (A media only control might be useful to show a lack of contamination in the collection media.)

Figure 2 shows the isolation plan – which is described as a bioassay guided fractionation – resulting in a single fraction, which contains a single compound (identified as pheophorbide A), that has the ability to induce MMP-2 and proMMP-9.  IT would be good to see a representative HPLC trace to show how the extract was fractionation by preparative HPLC.  It is not indicated how peaks were detected – optical density, CAD, or only MS.  Were other compounds present in the fractions – what was the cut off for identification and what was the percentage of the material comprised by the major, identified compound? (Could some of the bioactivity be caused by the presence of contaminating materials in the peak used for testing?)  Some of the MS and NMR data would be instructive in showing how the compounds were identified from the extracts and should be included (perhaps in a supplement?).

The data showing changes in MMP expression by zymography should be shown.  The results indicate that the differences in cleared area should by very obvious.  The PMA positive control is not very impressive for induction of MMP-9 secretion by HT1080 cells.  The figure shows error bars (n=3) but it is not indicated if this is from densitometry of independent zymograms – the source of the data needs to be described.

Table 1 shows the dose response for the PA on MMP production – again the zymogram would be useful and a description of how the data was obtained is required. The error bars are tiny considering the assay is presumably densitometry of a zymogram – an error of 2-5% of the average is not uncommon for densitometry of exactly the same gel. The table legend says other PA analog compounds were tested with “no significant differences detected” – differences from what?  The structures of the analogs are shown but the data of their activity is not shown – the data would be useful.

The authors addressed the possibility that PA caused MMP release into media following cell death by performing a cell viability assay however the specifics of this assay need to be described and a presentation of the data would be very useful.  Were the cells treated with multiple concentrations for several days?  Was this an MTT assay?

The HPLC method and apparatus could be more fully described.  For the step gradient, how long was each step set? I presume that portion #2 described in materials and methods is the same as FR #2 in Figure 2. Is portion #9 in materials described correctly? – should it be 100% ethyl acetate? Should portion #10 be 100% methanol?  The zymogram protocol indicates that the HT1080 cells were serum free for 3 days – were they still viable?

Reviewer 2 Report

Figure 1 – Have you done a quantitative analysis of the obtained results?

Figure 3 – There is only the quantitative analysis and gelatin zymography results were not shown. 

Was the activity of EA layer compared to that of single isolated compound?

Why did you choose the concentration 50microM? Have you determined the concentration of AQ20 in EA layer or tested it at the same concentration present in the leave extract?

Why did you choose HT1080 and did you test the activity of compounds only in one cell line?

Mtt cell viability assay was not described. 

Round 2

Reviewer 2 Report

The revised manuscript included all the revisions and could be accepted in present form.